# Seeds Priming with Melatonin Improves Root Hydraulic Conductivity of Wheat Varieties under Drought, Salinity, and Combined Stress

**DOI:** 10.3390/ijms25095055

**Published:** 2024-05-06

**Authors:** Yuanyuan Fu, Penghui Li, Zhuanyun Si, Shoutian Ma, Yang Gao

**Affiliations:** 1Institute of Farmland Irrigation, Chinese Academy of Agricultural Sciences, Xinxiang 453002, China; fyy2016060105@163.com (Y.F.); sizhuanyun@caas.cn (Z.S.); mashoutian@caas.cn (S.M.); 2Institute of Western Agriculture, Chinese Academy of Agricultural Sciences, Changji 831100, China

**Keywords:** melatonin, abiotic stress, priming effects, root hydraulic conductivity, aquaporin

## Abstract

Drought and salinity stress reduce root hydraulic conductivity of plant seedlings, and melatonin application positively mitigates stress-induced damage. However, the underlying effect of melatonin priming on root hydraulic conductivity of seedlings under drought–salinity combined remains greatly unclear. In the current report, we investigated the influence of seeds of three wheat lines’ 12 h priming with 100 μM of melatonin on root hydraulic conductivity (Lpr) and relevant physiological indicators of seedlings under PEG, NaCl, and PEG + NaCl combined stress. A previous study found that the combined PEG and NaCl stress remarkably reduced the Lpr of three wheat varieties, and its value could not be detected. Melatonin priming mitigated the adverse effects of combined PEG + NaCl stress on Lpr of H4399, Y1212, and X19 to 0.0071 mL·h^−1^·MPa^−1^, 0.2477 mL·h^−1^·MPa^−1^, and 0.4444 mL·h^−1^·MPa^−1^, respectively, by modulating translation levels of aquaporin genes and contributed root elongation and seedlings growth. The root length of H4399, Y1212, and X19 was increased by 129.07%, 141.64%, and 497.58%, respectively, after seeds pre-treatment with melatonin under PEG + NaCl combined stress. Melatonin -priming appreciably regulated antioxidant enzyme activities, reduced accumulation of osmotic regulators, decreased levels of malondialdehyde (MDA), and increased K^+^ content in stems and root of H4399, Y1212, and X19 under PEG + NaCl stress. The path investigation displayed that seeds primed with melatonin altered the modification of the path relationship between Lpr and leaf area under stress. The present study suggested that melatonin priming was a strategy as regards the enhancement of root hydraulic conductivity under PEG, NaCl, and PEG + NaCl stress, which efficiently enhanced wheat resistant to drought–salinity stress.

## 1. Introduction

Drought and salinity stress are two major causes for limited cereal crops development and growth from seed germination to harvest in many regions of the world [1,2,3].

Accordingly, drought and salt stresses give rise to several molecular, physiological, biochemical, and morphological changes in crop, such as up-regulated or down-regulated expression of relative genes, the accumulation of reactive oxygen species (ROS) and membrane osmotic regulators, ion toxicity, stomatal closure, the inhibition of photosynthesis, etc. [4,5,6,7,8,9,10]. Roots architecture, for its duty of absorbing and transporting water and nutrients, is pivotal to perceiving drought and salinity stresses. The well vigorous root system, conferring a drought/salinity-tolerant phenotype, could acquire sufficient water and nutrients, consequently increasing a larger leaf area and higher plant survival to improving crop production [11,12].

Currently, researchers use root hydraulic conductivity as an indication of water absorption and nutrient transport capacity [13,14]. The better the root system possesses the higher root hydraulic conductivity. Previous studies indicated that drought or salt stress reduced the root hydraulic conductivity in tomato, rice, zea, etc. [15,16,17,18,19,20]. And our recent research showed that the root hydraulic conductivity was reduced appreciably by combined drought and salinity stress in wheat varieties [21]. In addition, aquaporin genes (AQPs) regulate root hydraulic conductivity by controlling radially flowing intercellular transmembrane components in roots [22,23,24]. Studies have found that AQPs facilitated water and nutrition transport in numerous plant varieties [25,26,27,28,29,30,31,32]. Under drought and salt stresses, the expression and trafficking of AQPs were frequently regulated to control the water movement and played a critical role in the stress resistance [33,34,35,36]. In wheat, several TaAQPs were found and identified under drought and salt stresses. When wheat was subjected to drought and/or salt stress, many TaPIPs and TaTIPs were induced differential expression [37]. *TaAQP7* and *TaAQP8* were over-expressed in tobacco, resulting in the enhancement of resistance to drought and salt treatments [38,39]. *TaTIP2:2* modulated the response of *Arabidopsis thaliana* to drought and salt stresses independently of the abscisic acid metabolic pathway [40]. In the same manner, overexpression of *TaTIP4:1* in Arabidopsis and rice could heighten relative water contents to positively regulating seed germination and seedling growth under drought and salt stress [41]. Recently, researchers reported that the expression of *TaPIP1-2*, *PutTaPIP2-2*, *TaPIP2-4C1*, and *TaAQP2* in response to salt treatment was similar to day/night variation of root hydraulic conductivity [42]. Collectively, drought and salinity caused the reduction in root hydraulic conductivity, which was closely associated with AQPs expression. In addition, heightening root hydraulic conductivity could enhance plant resistance to drought, salinity, and drought–salinity combined stress.

Currently, diverse strategies have been applied to improve crop drought/salinity tolerance [43,44,45,46,47,48]. Seed priming as a seed treatment before sowing has been verified to be a way of assisting plants in defending against drought and/or salinity stresses via repairing cell membrane damage, enhancing seed vigor, actuating metabolic pathway, and affecting plant physiological and biochemical characteristics, etc. [49,50,51,52,53]. Melatonin, for its pleiotropic molecule role, regulated plants to resist drought/salinity stress by scavenging reactive oxygen species, promoting photosynthetic activity, and modulating a great number of metabolic pathways [54,55,56,57]. Furthermore, volumetric research works have demonstrated that seed priming with melatonin could improve drought and salt tolerance [58,59,60,61]. Melatonin as a priming agent promoted the formation of lateral root and seed germination under water stress [62]. Akbari et al. reported that seed priming using melatonin regulated oil content and fatty acid components of safflower under water deficit [63]. And in 2021, Heshmati et al. [64] implied that applying melatonin as a seed-priming agent improved safflower stability by keeping the membranes intact and antioxidant enzymes profession properly under drought condition. Zhang et al.’s [61] further research indicated that melatonin priming might improving cotton seedling tolerance through enhancing the scavenging system of active oxygen, aggravating the efficiency of photosynthesis, and coordinating to pathways of phytohormone signaling under salt stress. Latterly, studies denoted that seed priming with a suitable concentration of melatonin cloud increases root length, contents of soluble sugar, soluble protein, and proline, reduces malondialdehyde (MDA) content, and enhances antioxidant enzyme activities to alleviating the adverse effects of drought/salt stress [60,65,66]. Even so, the research addressing the seed priming with melatonin effect on root hydraulic conductivity of wheat seedlings under drought, salinity, and drought–salinity combined stress was limited.

Wheat (*Triticum aestivum* L.) is sensitive to drought stress and has moderate salt tolerance, providing staple food to about 40% of the world’s population [67,68]. Moreover, wheat cultivation was seriously affected by drought and salt stress [1,2]. However, research about the effects of melatonin on root hydraulic conductivity in wheat seedlings under drought, salt stress, and combined drought–salt stress is limited. In this study, similar to that of Fu et al. [21], seeds of H4399, Y1212, and X19 pre-treated with melatonin were germinated under polyethylene glycol 6000 (PEG-6000), NaCl, and PEG-NaCl combined stress. And these seedlings were treated with stress in hydroponics. Accordingly, we designed the present experiment to investigate the impact of seed priming with melatonin on root morphological structure, physiological traits, and gene expression of wheat seedlings under PEG, NaCl, and PEG-NaCl combined stress. The study testified the hypotheses: (1) melatonin priming could accelerate wheat seedling’s growth by improving the Lpr of three wheat varieties under the stress conditions, and (2) after seeds were triggered by melatonin, aquaporin genes regulated Lpr under PEG, NaCl, and PEG-NaCl combined stress. This study aimed to explore root hydraulic conductivity and its correlation with other physiological indicators of wheat seedlings after seeds were primed with melatonin.

## 2. Results

### 2.1. The Effects of Seed-Priming on Leaf Area and Root Length

From Figure 1, compared to the no-priming treatment, the leaf area of H4399 from water and melatonin priming was significantly increased under CK, PEG, and PEG + NaCl treatments, and that in melatonin-priming treatment increased more. Under NaCl stress, water-priming treatment remarkably reduced the leaf area of H4399 by 45.53%, and melatonin-priming treatment had no significant effect on the leaf area compared with no-priming treatment. In Y1212, relative to no-priming treatment, water-priming and melatonin-priming treatment strongly increased the leaf area under the CK condition and PEG + NaCl treatment. Under PEG and NaCl stresses, melatonin-priming treatment raised the leaf area under both conditions by 10.95% and 6.38%, respectively, and water-priming treatment insignificantly caused the change in leaf area compared to no-priming treatment. However, water-priming and melatonin-priming treatments considerably reduced the leaf area of X19 by 13.12% and 7.41% separately under the CK compared to no-priming treatment. Conversely, priming treatments drastically increased the leaf area of X19 under PEG, NaCl, and PEG + NaCl stresses, and that by melatonin-priming treatment showed more by comparison.

As shown in Figure 1, under CK condition, the root length of H4399 displayed obvious gains to 93.28% and 157.56%, respectively, by water-priming and melatonin-priming treatments, and it was increased more by melatonin-priming treatment compared to no-priming treatment. Under PEG stress, in H4399 this was increased significantly by 148.53% and 96.21% separately by water-priming and melatonin-priming treatments compared to no-priming treatment. Under NaCl stress, only melatonin-priming treatment gained strongly by 13.13% compared to no-priming treatment. Under PEG + NaCl stress, there was no significant difference in root length among three priming treatments. In Y1212, relative to no-priming treatment, water-priming and melatonin-priming treatments appreciably reduced root length under CK and NaCl conditions. Under PEG stress, water-priming treatment had no significant effect on root length, and melatonin-priming treatment diminished it by 4.85% compared with no-priming treatment. Similar to the change in root length in H4399 under PEG + NaCl stress, three-priming treatments had insignificantly influenced that in Y1212. Under PEG, NaCl, and PEG + NaCl stresses, water-priming and melatonin-priming treatments both drastically increased root length in contrast with no-priming treatment. The melatonin-priming treatment growth is more than that of the water-priming treatment under NaCl and PEG + NaCl stresses.

In addition, from Appendix A, compared to no-priming treatment, seed-priming treatments mostly considerably improved the root system of H4399, Y1212, and X19 under PEG, NaCl, and PEG + NaCl stresses, especially under PEG + NaCl stress. Furthermore, melatonin-priming treatment improved it more.

### 2.2. The Effects of Seed-Priming on Root Vitality

As shown in Figure 2, under CK condition, water-priming and melatonin-priming treatments appreciably depressed the root vitality of H4399 and X19 compared to no-priming treatment. Under PEG stress, by comparison, water-priming treatment had no significant effect on root vitality, and melatonin-priming treatment dramatically enhanced it by 55.76% in H4399. Whereas, seed-primed treatments markedly decreased root vitality of Y1212 and X19 under PEG stress, and melatonin-primed treatment reduced it less than water-primed treatment compared with no-primed treatment. Under NaCl stress, in H4399, relative to no-primed treatment, water-primed treatment and melatonin-primed treatment greatly reduced root vitality by 61.64% and 79.95% separately. Nevertheless, three-priming treatments did not affect root vitality of Y1212 and X19. Under PEG + NaCl stress, water-primed treatment and melatonin-primed treatment enhanced root vigor of H4399, Y1212, and X19, and melatonin-priming treatment raised it in Y1212 more relative to no-priming treatment.

### 2.3. The Effects of Seed-Priming on MDA Content

As shown in Figure 3, for the MDA content in shoot of H4399, water-primed and melatonin-primed treatments significantly reduced it under CK and PEG conditions by comparison. Under NaCl stress, there was no significant difference between no-priming and melatonin-priming treatments, and melatonin-priming treatment appreciably increased the MDA content in shoot by 31.11% compared with no-priming treatment. Obviously, under PEG + NaCl stress, the shoot of seedlings in seed-priming had the higher MDA content compared to no-priming treatment. In Y1212, two priming treatments extremely augmented the MDA content in shoot under CK, PEG, and NaCl states compared to the no-priming treatment. However, under PEG + NaCl stress, water-priming treatment significantly increased MDA content in shoot of Y1212, yet melatonin priming strongly reduced it compared to no-priming treatment. In X19, relative to no-priming treatment, water-priming and melatonin-priming treatments mostly caused a significant reduction in MDA content of shoot under CK, PEG, and NaCl conditions, yet a significant increase in it under PEG + NaCl stress. Under PEG stress, related to no-priming treatment, seeds priming strongly reduced MDA content in the root of H4399, and seeds priming with water caused a significant increase in it of Y1212 and X19. While there was no significant effect on MDA content in the root of Y1212, there was an obvious decrease in that of X19 under PEG stress by melatonin-priming treatment compared with no-priming treatment. Under NaCl stress, water-priming and melatonin-priming treatments appreciably reduced MDA content in the root of H4399 by 45.91% and 19.22%, respectively, in contrast. For MDA content in the root of Y1212 under NaCl stress, there was no difference among the three priming treatments. Notably, water-priming and melatonin-priming treatments had opposite effects on that of X19 under NaCl stress. Under PEG + NaCl stress, changes in the trend of seeds priming with water and melatonin on MDA content in the root of H4399 and Y1212 were similar to that under NaCl stress. Nevertheless, the water-priming treatment caused a significant increase, yet the melatonin-priming treatment a significant reduction in MDA content in the root of X19 under PEG + NaCl stress according to the no-priming treatment.

### 2.4. The Effects of Seed-Priming on Proline Content

As shown in Figure 4, in H4399, there was no significant difference in proline content in shoot among three priming treatments under CK and NaCl conditions. However, water-priming and melatonin-priming treatments dramatically reduced proline content in shoot under PEG and PEG + NaCl stresses (by 97.35% and 97.54% separately under PEG stress, by 78.58% and 82.60%, respectively, under PEG + NaCl stress) compared to no-priming treatment. In Y1212 and X19, the trends of proline content in shoot under CK, PEG, NaCl, and PEG + NaCl conditions were similar. Namely, water-priming and melatonin-priming treatments caused insignificant effect on proline content in shoot under CK, PEG, and NaCl states, yet a notable reduction in them under PEG + NaCl stress (by 79.94% and 93.88%, respectively, in Y1212; by 91.42% and 63.82% separately in X19) compared to no-priming treatment.

For proline content in root, water-priming and melatonin-priming treatments radically reduced them in H4399 (by 41.65% and 47.33% separately under CK condition, by 54.87% and 57.11% separately under PEG stress) under CK and PEG conditions relative to no-priming treatment. Under NaCl stress, there was no significant difference among three priming treatments in the content of H4399 by comparison. However, water-priming and melatonin-priming treatments markedly decreased the content of H4399 by 79.10% and 80.90% separately compared to no-priming treatment under PEG + NaCl stress. In Y1212 under CK and PEG states, water-priming and melatonin-priming treatments caused different trends of proline in root in contrast. Most notably, they induced a considerable increase under NaCl stress (by 97.11% and 41.81%, respectively), yet a significant reduction under PEG + NaCl stress (by 60.72% and 74.43% separately) for the content in root of Y1212 compared with no-priming treatment. In X19, in contrast, there was no difference between three priming treatments in proline content under CK, PEG, and NaCl conditions, while water-priming treatment extremely reduced the content in root by 38.36%, and melatonin-priming treatment greatly increased it by 90.43% under PEG + NaCl stress.

### 2.5. The Effects of Seed-Priming on Soluble Protein and Soluble Sugar

As can be seen from Figure 5A,C,E, water-priming and melatonin-priming treatments obviously increased soluble protein in shoots of H4399 and X19 under CK and PEG conditions compared to no-priming treatment. Under NaCl stress, there was no difference among the three priming treatments in soluble protein in the shoot of H4399. Nevertheless, water-priming and melatonin-priming treatments appreciably reduced soluble protein in shoots of H4399 by 49.20% and 35.46% separately under PEG + NaCl stress compared with no-priming treatment. In Y1212, water-priming and melatonin-priming treatments extremely reduced the content in shoot compared to no-priming treatment under CK, PEG, NaCl, and PEG + NaCl conditions. In X19, relative to no-priming treatment, only water-priming treatment markedly increased the content in shoot by 45.03% under NaCl stress, yet strongly reduced it by 23.63% under PEG + NaCl stress. Compared to no-priming treatment, water-priming and melatonin-priming treatments greatly reduced the content in roots of H4399 by 41.32% and 52.25% separately under CK condition. However, they had no influence on those of H4399 relative to no-priming treatment under PEG stress. Apparently, for soluble protein in roots of H4399, only water-priming treatment caused a significant increase under NaCl stress, yet melatonin-priming treatment also induced an increase under PEG + NaCl stress by comparison. In root of Y1212 and X19, compared to no-priming treatment, both melatonin-priming and water-priming treatments strongly reduced the content of soluble protein by 46.37% and 33.38% separately under CK condition. By comparison, water-priming treatment extremely reduced the content in roots of Y1212 by 38.46%, yet caused an insignificant change for that in X19 under PEG stress. However, melatonin-priming treatment did not obviously cause a reduction/increase in the root content of Y1212 and X19 under PEG stress. Under NaCl stress, relative to no-priming treatment, water-priming and melatonin-priming treatments insignificantly reduced the content in root of Y1212. Compared to no-priming treatment, water-priming treatment appreciably increased that by 31.51%, but melatonin-priming treatment remarkably decreased by 24.91% in X19 under NaCl stress. Under the PEG + NaCl condition, water-priming treatment extremely increased the content in roots of Y1212 by 30.53% but strongly reduced that of X19 by 33.57% in contrast with no-priming treatment, while there was no significant difference between no-priming and melatonin-priming treatments in root content of Y1212 and X19.

From Figure 5B,D,F, we can know the trend of soluble sugar. Compared to no-priming treatment, water-priming and melatonin-priming treatments markedly reduced the content in shoot of H4399 under CK, PEG, NaCl, and PEG + NaCl conditions. In shoot of Y1212, relative to no-priming treatment, two-priming treatment notably decreased the content under PEG and PEG + NaCl stresses. Nevertheless, by comparison, there was no appreciable difference between no-priming and two-priming treatments in the content under NaCl stress. In shoot of X19, compared to no-priming treatment, water-priming treatment extremely increased the content by 26.30% and 4.39% separately under CK and PEG conditions. Moreover, melatonin-priming treatment under PEG stress strongly reduced the content by 25.89% in comparison with no-priming treatment. Regrettably, in contrast, melatonin-priming treatment under NaCl significantly increased the content by 33.28%. Notably, two priming treatments reduced the content by 57.79% and 61.32%, respectively, under PEG + NaCl stress compared to no-priming treatment. For the content in the root, two priming treatments obviously reduced that of H4399 under PEG and PEG + NaCl stresses. However, compared to the no-priming group, two priming treatments markedly increased the content in the root of Y1212 under CK, PEG, and NaCl conditions, while water-priming and melatonin-priming treatments significantly reduced the content in Y1212 under PEG + NaCl stress. For the content in the root of X19, two priming treatments greatly increased it under CK, PEG, and PEG + NaCl conditions. Nevertheless, two priming treatments caused an insignificant increase in it under NaCl stress.

### 2.6. The Effects of Seed-Priming on Antioxidant Enzyme Activities

As shown in Table 1, for SOD activity in the shoot, water-priming and melatonin-priming treatments appreciably reduced that in H4399 and mostly reduced that of X19, yet increased that of Y1212 under CK, PEG, NaCl, and PEG + NaCl conditions compared to no-priming treatment. As for SOD activity in the root, in contrast, under CK condition, there was no significant difference between no-priming and two priming treatments in H4399. Under PEG stress, in comparison with no-priming treatment, water-priming treatment obviously increased SOD activity in the root of H4399, while two priming treatments markedly reduced that of H4399 under NaCl and PEG + NaCl stresses. In Y1212, by comparison, water-priming and melatonin-priming treatments both greatly increased SOD activity in roots under CK, PEG, NaCl, and PEG + NaCl conditions. In X19, melatonin-priming treatment extremely increased that in root under CK condition compared to no-priming treatment. Under PEG stress, in contrast with no-priming treatment, the changing trend of SOD activity in root caused by water-priming and melatonin-priming treatments was exactly the opposite. Under NaCl stress, only water-priming treatment caused a significant reduction in SOD activity in root by contrast with no-priming treatment. Under PEG + NaCl stress, two priming treatments remarkably increased SOD activity in the root of X19 by 387.70% and 582.00%, respectively, compared to the no-priming treatment.

From Table 1, under PEG and NaCl stresses, priming treatments caused no change in POD activity in shoots of H4399, while two-priming treatment under PEG + NaCl stress effected a significant increase in the activity in comparison with no-priming treatment. In the shoot of Y1212, in contrast, two priming treatments added POD activity under PEG stress by 78.81% and 110.59%, respectively, yet diminished the activity under PEG + NaCl stress. Under NaCl stress, only water-priming treatment strongly increased POD activity in the shoot of Y1212 by 29.97% compared to no-priming treatment. For X19, under PEG and PEG + NaCl stresses, in contrast, two priming treatments significantly reduced POD activity in the shoot. However, under NaCl stress, melatonin-priming treatment greatly increased the activity in the shoot by 67.03% compared to no-priming treatment. As for the POD activity in the root, by comparison, two priming treatments strongly added it in H4399 and X19 under CK condition. While under PEG stress, compared to no-priming treatment, two priming treatments only in X19 appreciably increased POD activity in root. Under NaCl stress, two priming treatments extremely decreased the activity in H4399 by 50.76% and 34.12%, respectively. Moreover, melatonin-priming treatment markedly reduced the activity in Y1212, yet increased that in X19 under NaCl stress in comparison with no-priming treatment. Only in H4399, two priming treatments under PEG + NaCl stress remarkably reduced POD activity in root in contrast with no-priming treatment.

As shown in Table 1, in shoot of H4399, in contrast, only water-priming treatment strongly reduced CAT activity by 30.41% under PEG stress, but only melatonin-priming greatly increased the activity by 35.00% under NaCl stress. However, two priming treatments greatly added the activity 84.35% and 56.50%, respectively, under PEG + NaCl stress compared to no-priming treatment. For the activity in root, by contrast with no-priming treatment, two priming treatments strongly reduced that of H4399 under PEG, NaCl, and PEG + NaCl stresses. From Table 1, two priming treatments greatly increased CAT activity in the shoot of Y1212 under CK, PEG, NaCl, and PEG + NaCl conditions compared to the no-priming treatment. As for CAT activity in the root, there was no significant difference between no-priming and two priming treatments under PEG stress. Notably, by contrast with no-priming treatment, water-priming and melatonin-priming treatments obviously increased the activity by 95.87% and 134.82% separately under NaCl stress. Additionally, under PEG + NaCl stress, only melatonin-priming treatment considerably reduced the activity by 26.76% compared to no-priming treatment. From Table 1, two priming treatments drastically reduced CAT activity in the shoot of X19 under CK and PEG conditions compared with no-priming treatment. In addition, under NaCl stress, only water-priming treatment remarkably reduced the activity by 12.50% compared to no-priming treatment. Nevertheless, water-priming and melatonin-priming treatments greatly increased the activity under PEG + NaCl stress by comparison, by 57.75% and 46.96%, respectively. For CAT activity in root, only melatonin-priming treatment strongly increased it under CK condition but reduced it under PEG stress by contrast with no-priming treatment. Additionally, two priming treatments radically increased it under NaCl and PEG + NaCl stresses compared to no-priming treatment.

### 2.7. The Effects of Seed-Priming on K^+^ and Na^+^ Content

As shown in Figure 6A,C,E, compared to no-priming treatment, two priming treatments significantly increased K^+^ content in the shoot of H4399 under PEG, NaCl, and PEG + NaCl stresses. In the shoot of Y1212, by contrast with the no-priming treatment, two priming treatments strongly increased K^+^ content under PEG and PEG + NaCl stresses. Under NaCl stress, only water-priming treatment greatly added the content by 10.58% compared with no-priming treatment. For K^+^ content in the shoot of X19, there was no significant difference between no-priming and two priming treatments under CK and NaCl conditions. Only water-priming treatment greatly increased the content by 5.52% under PEG stress. Furthermore, relative to no-priming treatment, water-priming and melatonin-priming treatments obviously increased the content by 34.68% and 32.88%, respectively, under PEG + NaCl stress. Additionally, compared to no-priming treatment, two priming treatments drastically increased K^+^ content in the root of H4399 under CK, PEG, NaCl, and PEG + NaCl conditions. In the root of Y1212, water-priming and melatonin-priming treatment strongly reduced K^+^ content by 4.57% and 9.86%, respectively, under PEG stress, yet increased the content by 58.28% and 79.12%, respectively, under PEG + NaCl stress compared to no-priming treatment. In addition, under NaCl stress, melatonin-priming treatment remarkably increased the content by 5.69% in compared with no-priming treatment. For K^+^ content in root of X19, melatonin-priming treatment considerably added it by 13.59% under PEG stress compared to no-priming treatment. Interestingly, there was no significant difference between no-priming and two priming treatments in K^+^ content under NaCl and PEG + NaCl stresses.

From Figure 6B,D,F, under the CK condition, two priming treatments had a less significant effect on Na^+^ content in shoots of H4399, Y1212, and X19 as compared to no-priming treatment. Compared to no-priming treatment, water-priming and melatonin-priming treatments greatly increased Na^+^ content in shoots of H4399 by 79.92% and 92.89% separately, yet had a less significant effect on the content of Y1212 and X19 under PEG stress. Under NaCl stress, water-priming and melatonin-priming treatments obviously reduced Na^+^ content in the shoot of H4399, but significantly increased the content of Y1212 and X19 in comparison with no-priming treatment. Under PEG + NaCl stress, two priming treatments strongly reduced Na^+^ content in shoots of H4399 and X19, yet extremely increased that of Y1212 compared to no-priming treatment. For Na^+^ content in the root of H4399, by comparison, two priming treatments markedly reduced it under PEG, NaCl, and PEG + NaCl stresses. In the root of Y1212, in contrast, water-priming and melatonin-priming treatments remarkably increased Na^+^ content under PEG, NaCl, and PEG + NaCl stresses. However, for Na^+^ content in the root of X19, water-priming and melatonin-priming treatments appreciably reduced it under PEG stress, yet obviously increased that under PEG + NaCl stress compared to no-priming treatment. Under NaCl stress, water-priming treatment greatly reduced the content by 11.33%, but melatonin-priming treatment had a less significant effect on the content compared with no-priming treatment.

### 2.8. The Effects of Seed-Priming on Root Hydraulic Conductivity and Theoretical Hydraulic Conductivity

We analyzed the root canal tissue under PEG + NaCl stress. From Figure 7, compared to no-priming treatment, water-priming and melatonin-priming treatments obviously increased root canal tissue of H4399, Y1212, and X19 under PEG + NaCl stress.

In addition, as shown in Figure 8A,C,E, compared to no-priming treatment, water-priming treatment extremely increased root hydraulic conductivity of H4399 under PEG and PEG + NaCl stresses. By contrast, water-priming treatment greatly increased root hydraulic conductivity of Y1212 only under PEG + NaCl stress. Moreover, for that in X19, relative to no-priming treatment, water-priming treatment strongly increased it under CK, PEG, and PEG + NaCl conditions. Obviously, compared to no-priming treatment, melatonin-priming treatment significantly increased root hydraulic conductivity of H4399, Y1212, and X19 under CK, PEG, NaCl, and PEG + NaCl conditions.

From Figure 8B,D,F, compared to no-priming treatment, water-priming treatment significantly increased theoretical hydraulic conductivity of H4399 under PEG, NaCl, and PEG + NaCl stresses. For that in Y1212, by comparison, water-priming treatment strongly increased it under CK, PEG, NaCl, and PEG + NaCl conditions. However, in X19, by comparison, water-priming treatment greatly increased theoretical hydraulic conductivity only under CK conditions. Expectantly, compared to no-priming treatment, melatonin-priming treatment mostly greatly increased theoretical hydraulic conductivity of H4399, Y1212, and X19 under CK, PEG, NaCl, and PEG + NaCl conditions.

### 2.9. The Effects of Seed-Priming on Expression of Some Aquaporin Genes in Roots

From Figure 9, compared to no-priming treatment, water-priming and melatonin-priming treatments up-regulated/down-regulated aquaporin genes in H4399, Y1212, and X19 under CK, PEG, NaCl, and PEG + NaCl conditions. In H4399, by comparison, water-priming and melatonin-priming treatments greatly up-regulated the expression levels of these genes under CK, PEG, NaCl, and PEG + NaCl conditions. In H4399, by comparison, the up-regulated expression level of *TaSIP2;02_4A* in water-priming treatment was highest under PEG stress, and that in melatonin-priming treatment was highest under NaCl stress. For the expression level of *TaNIP3;03_6D*, in water-priming treatment, it was highest under PEG + NaCl stress. In addition, it was highest in melatonin-priming treatment under NaCl stress. As for transcript levels of *TaNIP2;04a_7D*, in melatonin-priming treatment it was highest under PEG + NaCl stress. In Y1212, by comparison, the transcript levels of *TaNIP3;03_6D* in water-priming treatment were strongly elevated under CK, PEG, NaCl, and PEG + NaCl conditions, and its up-levels were highest under PEG stress. The transcript levels of *TaPIP1;01_2D* were greatly reduced in water-priming treatment under PEG + NaCl stress by comparison. Melatonin-priming treatment down-regulated the expression of most of these genes under PEG, NaCl, and PEG + NaCl stresses in comparison with no-priming treatment. In X19, compared to no-priming treatment, water-priming and melatonin-priming treatments down-regulated the expression levels of *TaNIP3;03_6D* under CK, PEG, and NaCl conditions. Among all treatments, *TaNIP1;08_7D* and *TaPIP1;01_2D* in X19 showed increased expression levels in water-priming treatment under PEG + NaCl stress by comparison.

### 2.10. The Effects of Seed-Priming on the Relation between Root Hydraulic Conductivity and the Physiological Characteristics

As shown in Figure 10, the results of correlation analysis among the three priming treatments were almost similar but various. Lpr had a significantly positive association with characteristics like RL, La, Ks, and Kr and a strongly negative association with characteristics like Nas, Nar, Pros, Pror, Sus, Sur, SODs, SODr, PODs, and CATr among no-priming, water-priming, and melatonin-priming treatments. During no-priming and water-priming treatments, Lpr was significantly and passively correlated with MDAr and PODr. Differently, Lpr had a greatly positive relationship with CATs during no-priming treatment but with Sps during water-priming treatment. Lpr had a negative association with Sps among no-priming treatment but with MDAs and CATs among water-priming treatment and with Ra among melatonin-priming treatment. For TLpr, it was significantly and actively correlated with traits like RL, La, Ks, and Kr, while it was significantly and passively correlated with traits like Nas, Nar, Pros, Sus, and PODs among no-priming, water-priming, and melatonin-priming treatments. TLpr had a strongly positive association with CATs among no-priming and melatonin-priming treatments. In addition, TLpr had a negative association with traits like Pror, PODr, and CATr among no-priming and water-priming treatments. Among no-priming treatment, the traits which had a significantly positive association correlated with TLpr were MDAr, SODs, and SODr. TLpr was obviously and passively correlated with Sur among water-priming treatment but with Spr among melatonin-priming treatment.

Furthermore, the path analysis was used for effect of seeds priming on the relationship between Lpr and all these variable. As shown in Appendix A, in no-priming treatment, RL greatly affected many traits, such as SODr, La, Lpr, Sus, and Pros. However, in water-priming treatment, RL only influenced La. Differently, Ks had a direct significant association with Lpr, and the value of correlation was 0.952 and 0.476 in no-priming and water-priming treatments, respectively, while Lpr was not strongly and directly influenced by Ks in melatonin-priming treatment. In no-priming treatment, La positively and greatly influenced Lpr. Yet, in water-priming and melatonin-priming treatments, La was positively and greatly influenced by Lpr, and their correlation coefficient values reached 0.322 and 0.799, respectively.

## 3. Discussion

### 3.1. Seed-Priming Using Melatonin Increasing Root Hydraulic Conductivity of Wheat Seedlings

Roots are the first and most important organ to sense and resist drought and salt stresses. Among root physical parameters, root hydraulic conductivity (Lpr) is a representative index, controlling hydraulic conductance and determining plant vigor under drought and salt stress [14]. Recently, it was shown that drought stress, salt stress, and dual drought–salt stress obviously reduced Lpr [19,21,69]. And root length/leaf area positively and significantly affected Lpr of Hengmai 4399, Yannong 1212, and Xinmai 19 under PEG and salt stress [21]. In this experiment, seed priming strongly improved root hydraulic conductivity of H4399, Y1212, and X19 under dual PEG and salt stress to enhance root water and ion uptake capacity (Figure 8). In addition, melatonin seed priming dramatically ameliorated it under single PEG and single salt stresses in H4399, Y1212, and X19 (Figure 8), indicating that melatonin priming was more effective than water priming in improving Lpr.

Several studies found that root hydraulic conductance was regulated by aquaporins and melatonin increased the translation level of PIPs (plasma membrane intrinsic proteins) to enhance root hydraulic conductance in rice and maize, which were one of the aquaporins subfamilies [70,71,72]. Our results showed that water and melatonin seed priming modulated the expression of some genes in the NIPs (NOD26-like intrinsic proteins), PIPs, TIPs (tonoplast intrinsic proteins), and SIPs (small basic intrinsic proteins) subfamily under PEG, NaCl, and PEG + NaCl stresses (Figure 9), which pertained to aquaporin’s subfamily. Additionally, the translation levels of these genes modulated by water-priming and melatonin-priming treatments differed among H4399, Y1212, and X19 (Figure 9), indicating that there were differences in the regulation of root hydraulic conductivity by aquaporins in different wheat varieties. In addition, the translation levels of aquaporins in one wheat variety from melatonin priming under stresses were not all enhanced, which may be relative to the sampling time, function of aquaporin’s subfamily genes, etc.

Furthermore, correlation analysis in the present research shows that Lpr was appreciably associated with most traits under seed-priming treatments (Appendix A), so the higher Lpr may be the result of more active antioxidant activities, lower cell membrane damage, less osmotic regulatory substance, and higher inorganic ions caused by seed priming under PEG + NaCl stress. Moreover, the findings of path analysis revealed that water seed priming and melatonin seed priming led to the modification of the path relationship between Lpr and La (Figure 10). Leaf area was a manifestation of crop yield and was closely related to photosynthetic efficiency [73]. Regrettably, photosynthetic parameters of seedlings from no-primed, water-primed, and melatonin-primed treatments under CK, PEG, NaCl, and PEG + NaCl conditions were not measured. Thereby, the association between Lpr and photosynthetic by correlation of leaf area was absent, which needed to be precisely studied.

### 3.2. Seed-Priming Using Melatonin Promoting Wheat Growth

Leaf area, root length, and root activity are important parameters of plant growth under abiotic stress [74,75]. In our study, under PEG + NaCl stress, water-priming and melatonin-priming treatments significantly increased leaf area and root activity of three wheat varieties and root length of X19 to promoting plant growth compared to no-priming treatment (Figure 1 and Figure 2), which were similar to these findings of Khan et al. and Wei et al. [59,65]. However, the root length of H4399 and Y1212 between no-priming treatment and priming treatments did not differ under PEG + NaCl stress, indicating that there were differences in the response of different wheat varieties to drought and salt dual stress. Under single and combined PEG and NaCl stresses, water and melatonin priming significantly increased leaf area and root length of X19 compared to no-priming treatment (Figure 1), implying that X19 seedlings might have higher drought resistance and salt tolerance after seed-priming using water and melatonin. Additionally, as a whole, the effect of melatonin being applied as a seed-priming agent on leaf area and root length of three wheat seedlings was superior to water as a priming agent under CK, PEG, NaCl, and PEG + NaCl conditions.

Previous studies showed that melatonin priming increased contents of MDA, proline, soluble sugar, and soluble protein to improve salt or drought tolerance of seedlings [65,66]. However, in the present study, by comparison, water-priming and melatonin-priming treatments reduced levels of proline, soluble protein, and soluble sugar in three wheat varieties to accelerate plant growth (Figure 4, Figure 5 and Figure 6), and melatonin priming reduced MDA content in the root of H4399 and X19 (Figure 3). Similar findings were observed in tomato, wheat, maize, and rapeseed under drought stress [59,60,76,77]. Reports revealed that melatonin decreased the proline content under drought stress [78], similar to our findings, while water priming greatly increased MDA content in the shoot of H4399, Y1212, and X19 compared to no priming, indicating that melatonin priming decreased cell membrane damage more than water priming and more effectively enhanced seedlings resistance to drought and salt combined stress for H4399, Y1212, and X19.

Specifically, melatonin and water priming significantly increased K^+^ content in root and shoot of three wheat varieties under PEG + NaCl stress by contrast (Figure 6). Interestingly, water-priming and melatonin-priming treatments obviously reduced Na^+^ content in the root and shoot of H4399 but increased it in the root and shoot of Y1212 and X19 under PEG + NaCl stress compared to no-priming treatment (Figure 6). Previous studies showed that melatonin priming promotes K^+^ content and reduces Na^+^ content under salt stress [79]. Our results showed that the reduction of Na^+^ content caused by melatonin priming was also related to different varieties. In addition, we speculated that three wheat varieties with higher inorganic ions might improve the ability of seedlings to withstand ion toxicity, and the contents did not hinder seedlings’ growth and function under drought–salt dual stress after seeds priming with water and melatonin.

### 3.3. Seed-Priming Inducing Stress Responses

Previous researchers showed that abundant metabolic processes were triggered after seeds priming to enhancing seedling resistance to stress [51,80]; therefore, because of ‘priming memory’, primed seeds owned the increase of stress tolerance. Presumably, the response of seedlings to stress after priming may be epigenetically regulated, facilitating molecular knowledge of ‘priming memory’. DNA methylation and histone modification, altering levels of gene expression, had been proposed as a key point to impart ‘priming memory’ and the increase of stress tolerance [52,81,82,83].

Currently, it is not completely confirmed about the epigenetic control mechanism that induces the regulation of metabolic processes [84]. Nevertheless, some results of seed germination implied possible targets of the regulation. For instance, during seed germination, cytosine methylation modulates gene silencing [81,85], while histone deacetylase (HDAC) regulates tolerance of osmotic/salt stress of Arabidopsis seeds through increasing late embryogenesis abundant (LEA) because of over-expression of *AtHD2C* [81,86]. Yuan et al. [87] showed that histone acetylation/deacetylation was an epigenetic regulatory mechanism of chromatin structure in plant development and stress, which modulated the cell cycle and maintained genome stability. In addition, Kubala et al. [80] found that transcription levels of *HAC7* encoding histone acetyltransferase were up-regulated in osmoprimed B.napus seeds. It is of great significance to delve into the epigenetic aspects of seed priming.

In addition, in this study, it seems that H4399 showed a most positive effect on seed priming, such as lower levels of MDA and proline, higher contents of K^+^, and less Na^+^ content under PEG + NaCl stress (Figure 3, Figure 4, and Figure 6). We speculated that this was mainly related to its drought and salt tolerance [88,89]. And higher stress tolerance of H4399 may be the result of specific epigenetics.

In conclusion, our study corroborated that melatonin priming had a wonderful role in improving root hydraulic conductivity, regulating antioxidant enzyme activity, and reducing cell membrane damage, eventually leading to promoting root growth and plant development under PEG, NaCl, and PEG + NaCl stresses. In addition, as shown in Figure 11, according to the morphological and physiological response of seedlings to the dual stress of PEG and NaCl, we recapitulated the response in three wheat varieties. The present study promoted the understanding of water-primed and melatonin-primed seedlings’ resistance to dual PEG and salt stress, especially in the root hydraulic conductivity. However, further studies are needed to explore the effect of melatonin on root hydraulic conductivity under abiotic stress.

## 4. Materials and Methods

### 4.1. Plant Materials

In the present experiment, Heng4399 (H4399, resistance to drought and salt stress), Yannong1212 (Y1212, moderately resistance to drought stress and sensitive to salt stress), and Xinmai19 (X19, sensitive to drought and salt stress) were acquired from the Dry-Land Farming Institute of Hebei Academy of Agricultural and Forestry Sciences and used for treating, according to Fu et al. [21]. The trial was organized in a controlled environmental chamber at the Experimental Station of Farmland Irrigation Institute (35°54′ N, 113°29′ E, and 80.77 m altitude), located in Qiliying, Xinxiang, Henan, China. The environmental controls of the chamber contained the day/night temperature of 25 °C/20 °C, the 12 h photoperiod of 600 μmol·m^−2.^s^−1^ from 7:00 to 19:00, and the relative humidity of 40–50%.

### 4.2. Experimental Design

We set up three seed priming and four stress treatments, with fifteen replicates per treatment group. The three seed primings included seed no-priming (N), seed water-priming (W), and seed melatonin-priming (M). A concentration of 100 μM melatonin was applied, according to the results of the previous research [90]. In addition, four stress treatments consisted of PEG-induced drought stress (2% PEG 6000, PEG), Salt stress (0.1% NaCl, NaCl), drought–salt combined stress (2% PEG 6000 and 0.1% NaCl, PEG + NaCl), and the untreated group (CK). During treatment, Hoagland solution was used. In accordance with Li et al. [91], 1 L of Hoagland solution included 0.506 g of KNO_3_, 1.181 g of Ca(NO_3_)_2_ 4H_2_O, 0.136 g of KH_2_PO_4_, 0.246 g of MgSO_4_·7H_2_O, and 1 mL of micronutrients (23 g/L of C_10_H_12_FeN_2_NaO_8_·3H_2_O, 2.86 g/L of H_3_BO_3_, 1.55 g/L of MnSO4·H_2_O, 0.2 mg/L of ZnSO_4_·7H_2_O, 0.08 g/L of CuSO4·5H_2_O, 0.09 g/L of H_2_MoO_4_, pH = 6.0).

We selected uniform size and full seeds and soaked them under 4 °C darkness conditions for priming. The time of soaking seeds in distilled water and melatonin was 12 h, and then these seeds were naturally dried until initial seed water content. Afterwards, we conducted seed germination and seven-day seedling treatments according to Fu et al. [21]. We used 9 cm diameter petri dishes to place seeds, which contain four layers of filter paper impregnated with different treatment solutions. In a phytotron, these seeds were incubated at 25 °C of darkness. After 2-day incubation, light was applied. Then 12 cm × 30 cm of PVC solution culture barrels were used to hydroponic 7-day seedlings in a chamber. After seventeen days, we harvested the seedlings in all treatments to measure all indicators.

### 4.3. Determination of Root Growth and Leaf Area

We used an Epson V800 root scanner (Perfection V800, Shanghai, China) to scan the roots of H4399, Y1212, and X19 under different treatments and observe the root structure. WinRHIZO 7.4.2 software (Rengent Instruments Inc., Montréal, Québec, Vancouver, Canada) was used to measure root length.

We used the equation (length × width of leaf × 0.85) to calculate the leaf area of fully unfolded leaves.

### 4.4. Measurem Ent of Root Viability

We cut the roots of five wheat plants into about 1 cm of small segments. Then, 0.3 g sample tissues were weighed and put into a centrifuge tube. Next, 5 mL of 0.4% triphenyltetrazolium chloride (TTC) and 0.2 mol/L phosphate buffer (ph = 7.0) were added to the tube. After completely immersing root tissues in the reaction solution, we incubated the tube at 37 °C for 3 h in an incubator to make the severed root tip red. Then, we immediately terminated the reaction with 1 mol/L of H_2_SO_4._ We took out the stained root tissues and put them into 20 mL of methanol for 4 h. Finally, the supernatant was colorimetric-analyzed at 485 nm. The root vitality was calculated by using the reduction amount of tetrazolium [92].

### 4.5. Measurement of Relative Conductivity in Shoot

We analyzed the relative conductivity in shoot. First, 0.1 g of fresh sample was weighed, and 10 mL of deionized water was added. Then, a conductivity meter (SX723, Thermo, American) was used for measuring the conductivity value (R1) after standing for 24 h. Then, it was boiled at 100 °C for 30 min in a constant-temperature water bath (HH.S21-8, Boxun, Shanghai, China), and we measured the value (R2) after cooling. The value of relative conductivity was calculated by the equation (R1/R2 × 100%).

### 4.6. Measurement of Root Hydraulic Conductivity and Theoretical Hydraulic Conductivity

According to Fu et al. [21], the root hydraulic conductivity (Lpr) and theoretical hydraulic conductivity (TLpr) were analyzed using Lpr = Jv/ΔP and $K_{xylem}$=$\frac{\pi }{128ƞ}\sum_{i=1}^{n}D_{i}^{4}$, respectively [91,93,94].

For Lpr, a pressure chamber (Model 3115, Plant Moisture Equipment, Santa Barbara, CA, USA) was used for measuring it. Jv indicates the slope of the water flux, and ΔP indicates the pressure difference curve.

As for TLpr, the roots of seedlings were rinsed with deionized water, and root tips were cut into a size of 0–1 cm. Then, the tissues were quickly placed in a formaldehyde–acetic acid–ethanol fixative (FAA) solution for 48 h. After, they performed routine paraffin sectioning by staining with saffron green. The rotary slicer (RM2235, Leica, Germany) was used for slicing them at a thickness of 10 µ m. The tissues were observed through a digital photography microscope (Axiolab A1, ZEISS, Oberkochen, Germany). ImageJ 32 measured the root diameter (µ m) and duct diameter (D, µ m). In the Hagen–Poiseuille equation, ƞ represents the viscosity coefficient of water, and the value of it is 0.90 × 10^−6^ KPa. The n indicates conduit number. In addition, the root water potential was the pressure in the pressure chamber when Lpr was measured.

### 4.7. Detection of Antioxidant Enzymes Activity, Levels of Malondialdehyde, Contents of Osmoregulatory Substances, and Amounts of Potassium and Sodium

Following Fu et al. [21], the activities of antioxidant enzymes (SOD, POD, and CAT) were measured by using kits (Comin Biotechnology Co., Suzhou, China), including SOD-2-W, POD-2-Y, and CAT-1-W. Following the kits’ instructions, the fresh samples of 0.5 g were added into a centrifuge tube containing the crude enzyme extract. After grinding and breaking, it was centrifuged at 10,000× *g* for 10 min at 4 °C in a centrifuge (Cence, TGL-20M) to obtain the supernatant. Then, the supernatant was used for measuring activities of antioxidant enzymes at wavelengths of 450 nm, 470 nm, and 405 nm, respectively.

The contents of malondialdehyde (MDA) and osmoregulatory substances (proline, soluble sugar, and soluble protein) were measured by the kits, including MDA-2-Y, PRO-2-Y, KT-2-Y, and BCAP-2-W. The wavelengths of 532 nm and 600 nm were determined to measure content of MDA. The wavelengths of 520 nm, 562 nm, and 620 nm were measured contents of proline, soluble sugar, and soluble protein, respectively.

Contents of sodium (Na^+^) and potassium (K^+^) were analyzed through a flame photometer (FP650, Shanghai, China) following the operating instructions of Fu et al. [21]. The 0.15 g of dry samples of wheat seedlings was used and dissolved in H_2_SO_4_-H_2_O_2_. After, the solution was used for determining the levels of Na^+^ and K^+^.

### 4.8. qPCR Analysis

We collected roots from wheat seedlings by using liquid nitrogen and stored them subsequently at −80 °C. Their total RNA was extracted from roots utilizing a Hipure Plant RNA Mini Kit (Magen, Guangzhou, China). Then, first-strand cDNA synthesis and qPCR analysis were performed following Li et al. [95]. *TaActin* gene in wheat was applied as an endogenous control. Primers used in the qPCR are listed in Appendix A. The 2^−∆∆CT^ algorithm was used for calculating the relative gene expression.

### 4.9. Statistical Analysis

Excel 2021 (Microsoft, USA), Origin 2021 (Origin Lab, Northampton, MA, USA), and SPSS 20.0 (IBM Corp., Armonk, Chicago, USA) were used for basic data statistics, data processing, and data drawing, respectively. The results were presented as mean values±SDs. We used Duncan’s method at the level of *p* < 0.05 to determine the significant differences. The R 4.2.2 and RStudio software 2023.06.1 were used to draw the heat map. The correlation between traits was analyzed using Pearson correlation coefficient (r). The path analysis in this study was executed by SPSSPRO (http://www.spsspro.com, accessed on 12 December 2023).

## 5. Conclusions

Collectively, melatonin application as a seed priming reagent effectively modulated the transcription level of aquaporin genes and improved root hydraulic conductivity in H4399, Y1212, and X19 under single PEG, single NaCl, and dual PEG + NaCl stresses. Melatonin seed priming regulated antioxidant enzyme activity, which highlighted the essential role of melatonin in the removal of reactive oxygen and enhancing wheat resistance. Furthermore, melatonin-primed treatment reduced cell membrane damage, alleviated accumulation of osmosis substances, increased potassium absorption, and inhibited efflux under PEG + NaCl stress. These behaviors indicated that melatonin priming aggravated the ability of wheat seedlings against PEG and NaCl combined stress, eventually promoting root elongation and leaf growth of seedlings. The findings in the present report assisted in enhancing the apprehension of the effect of melatonin priming on root hydraulic conductivity against single PEG, single NaCl, and dual PEG + NaCl stresses.

## Figures and Tables

**Figure 1 ijms-25-05055-f001:**
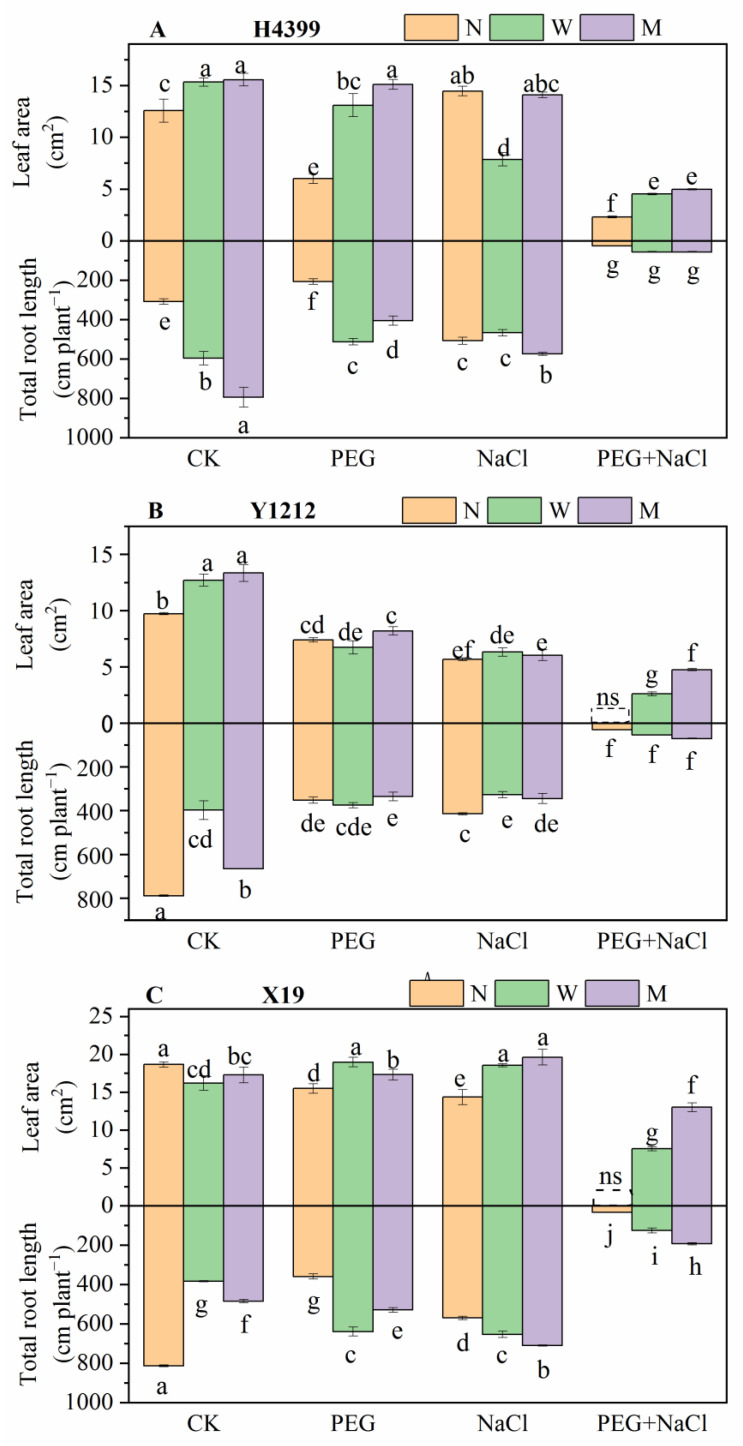
The leaf area and root length of three wheat seedlings from no priming (N), water priming (W), and melatonin priming (M) under PEG, NaCl, and PEG + NaCl treatments. The standard deviation (SD) value from three replicates is represented as error bars in figures, and different lowercase letters indicate significant differences at the 0.05 level.

**Figure 2 ijms-25-05055-f002:**
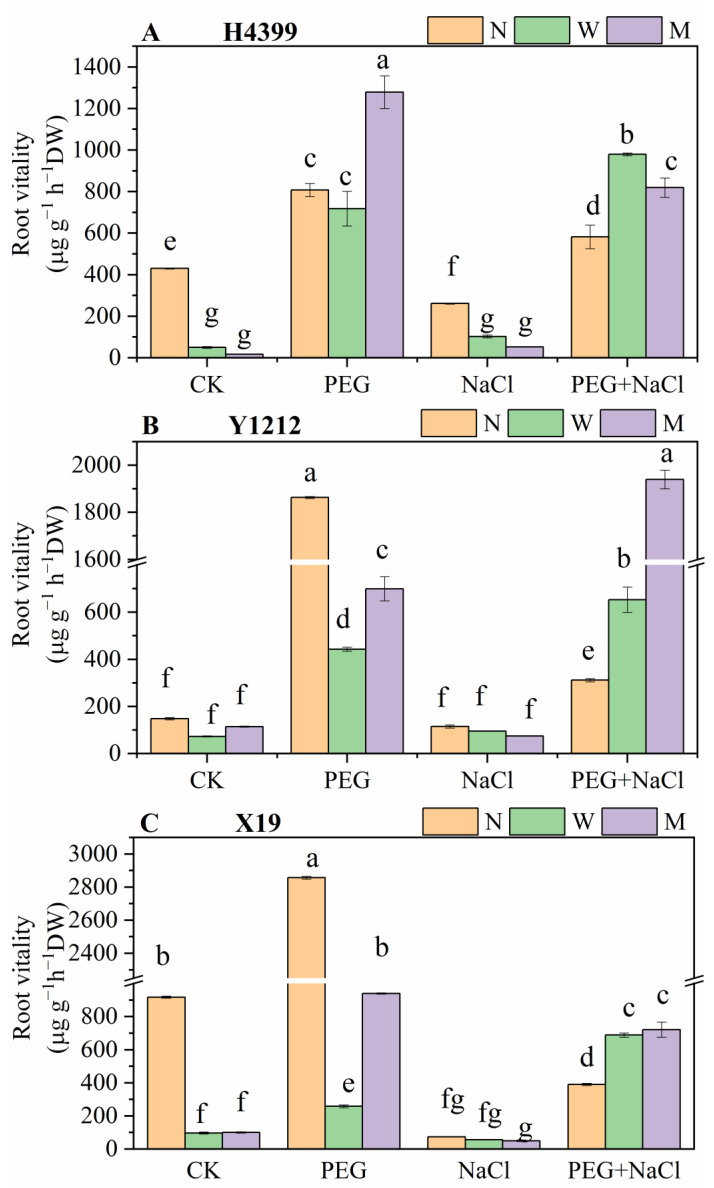
The root vitality of three wheat seedlings from no priming (N), water priming (W), and melatonin priming (M) under PEG, NaCl, and PEG + NaCl treatments. The standard deviation (SD) value from three replicates is represented as error bars in figures, and different lowercase letters indicate significant differences at the 0.05 level.

**Figure 3 ijms-25-05055-f003:**
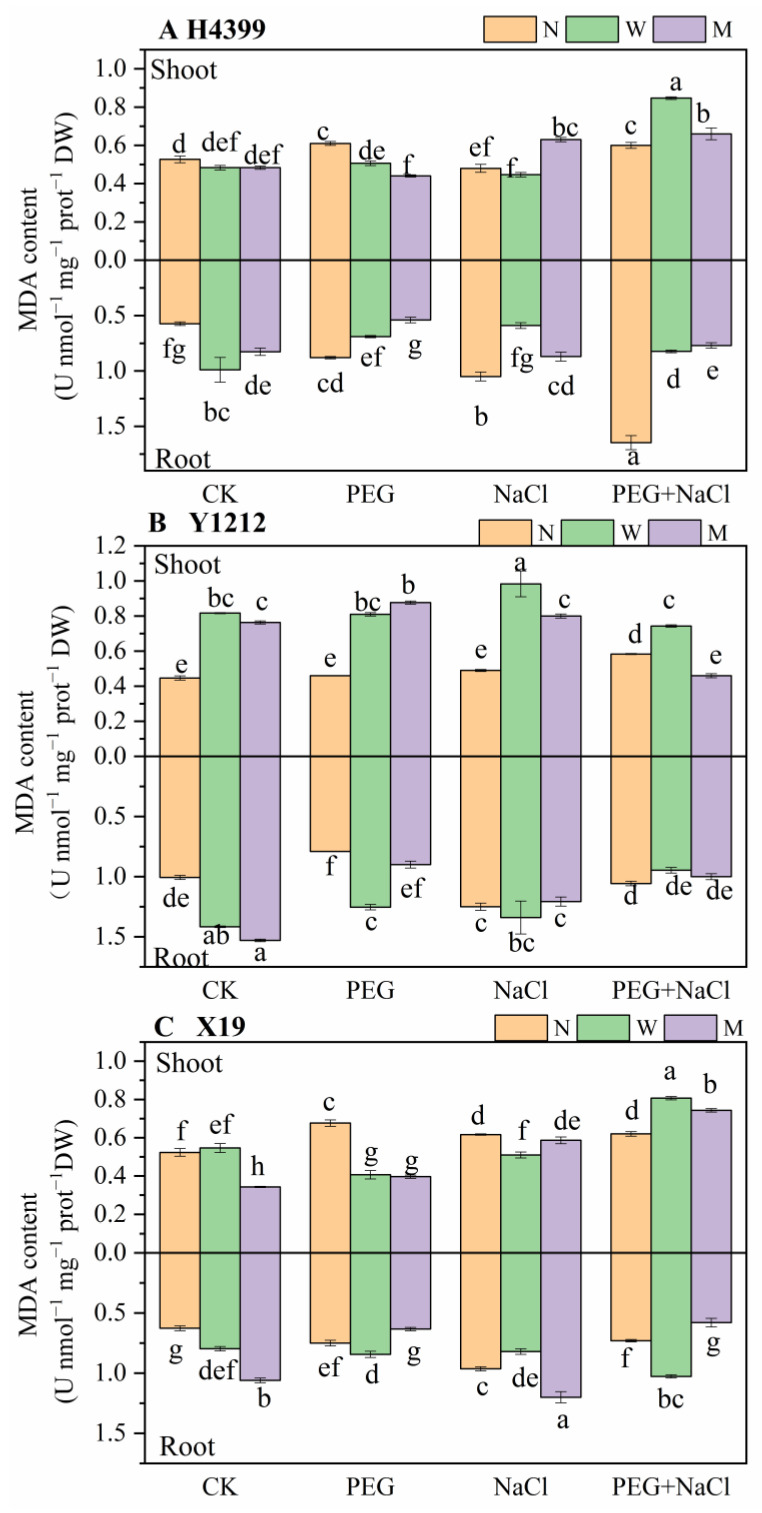
The MDA content in shoot and root of three wheat seedlings from no priming (N), water priming (W), and melatonin priming (M) under PEG, NaCl, and PEG + NaCl treatments. The standard deviation (SD) value from three replicates is represented as error bars in figures, and different lowercase letters indicate significant differences at the 0.05 level.

**Figure 4 ijms-25-05055-f004:**
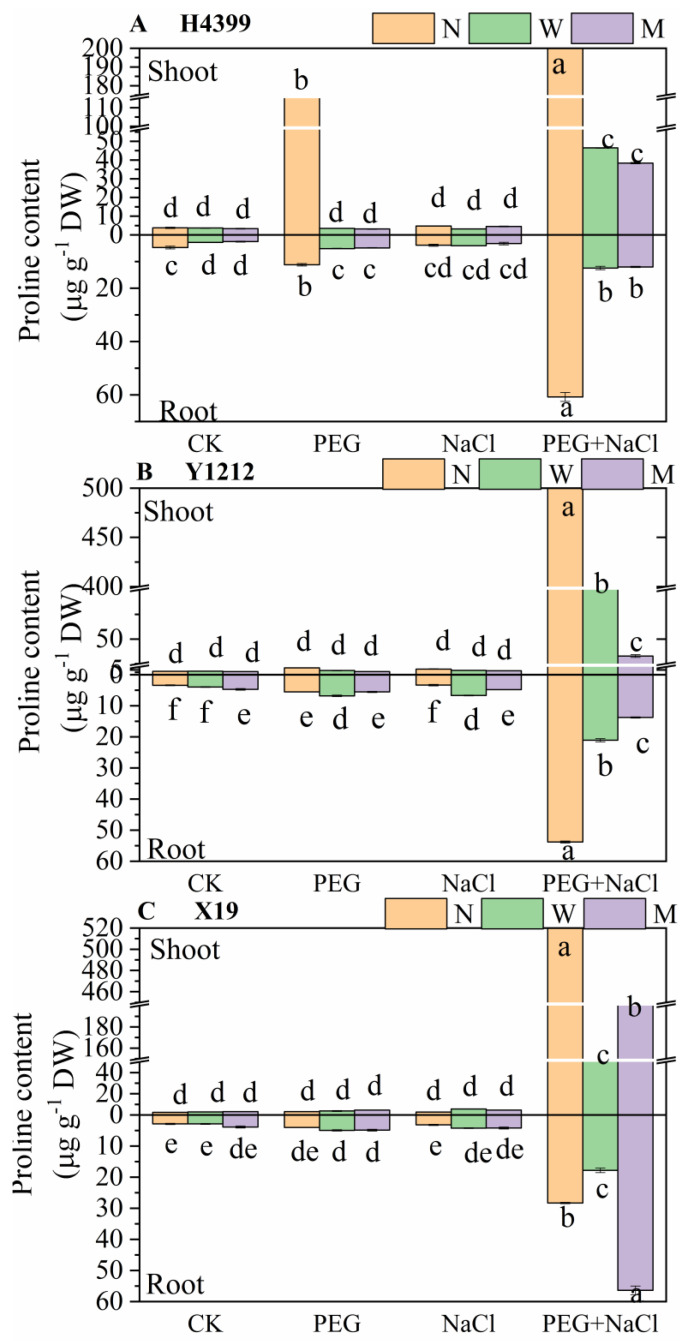
The proline content in shoot and root of three wheat seedlings from no priming (N), water priming (W), and melatonin priming (M) under PEG, NaCl, and PEG + NaCl treatments. The standard deviation (SD) value from three replicates is represented as error bars in figures, and different lowercase letters indicate significant differences at the 0.05 level.

**Figure 5 ijms-25-05055-f005:**
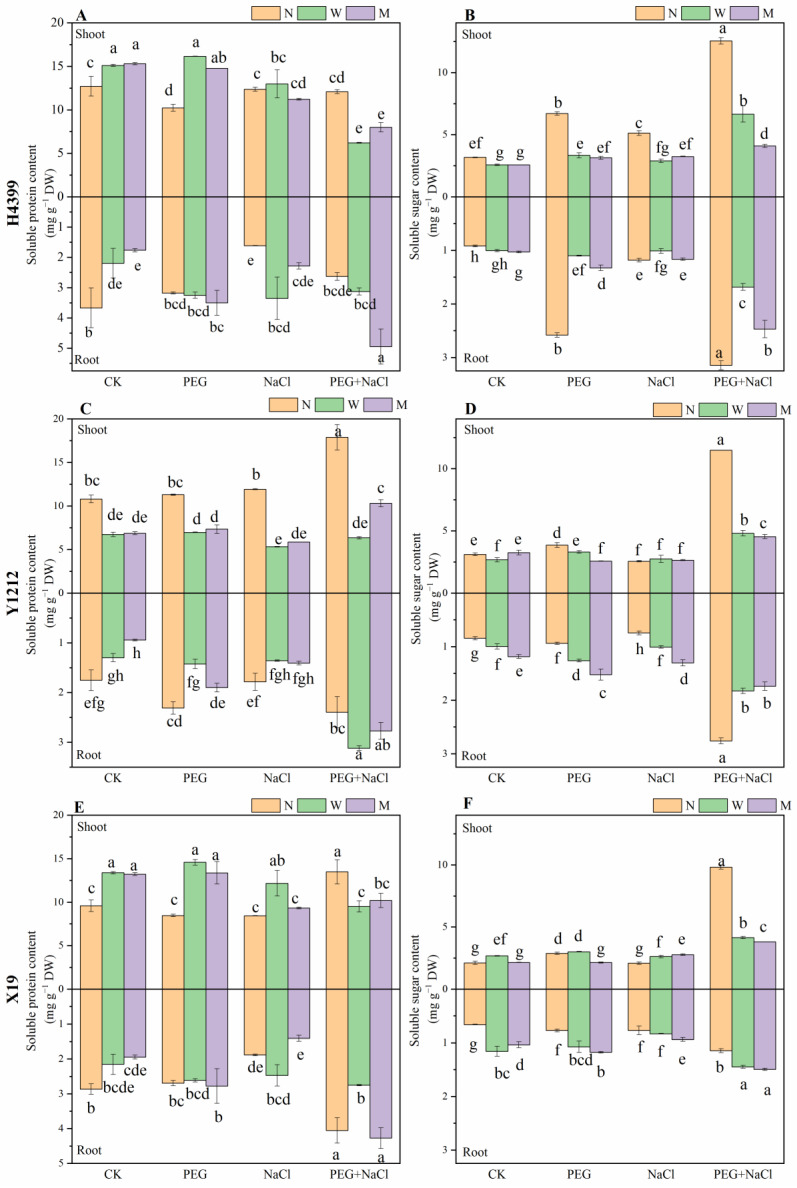
The soluble protein and soluble sugar in shoot and root of three wheat seedlings from no priming (N), water priming (W), and melatonin priming (M) under PEG, NaCl, and PEG + NaCl treatments. The standard deviation (SD) value from three replicates is represented as error bars in figures, and different lowercase letters indicate significant differences at the 0.05 level.

**Figure 6 ijms-25-05055-f006:**
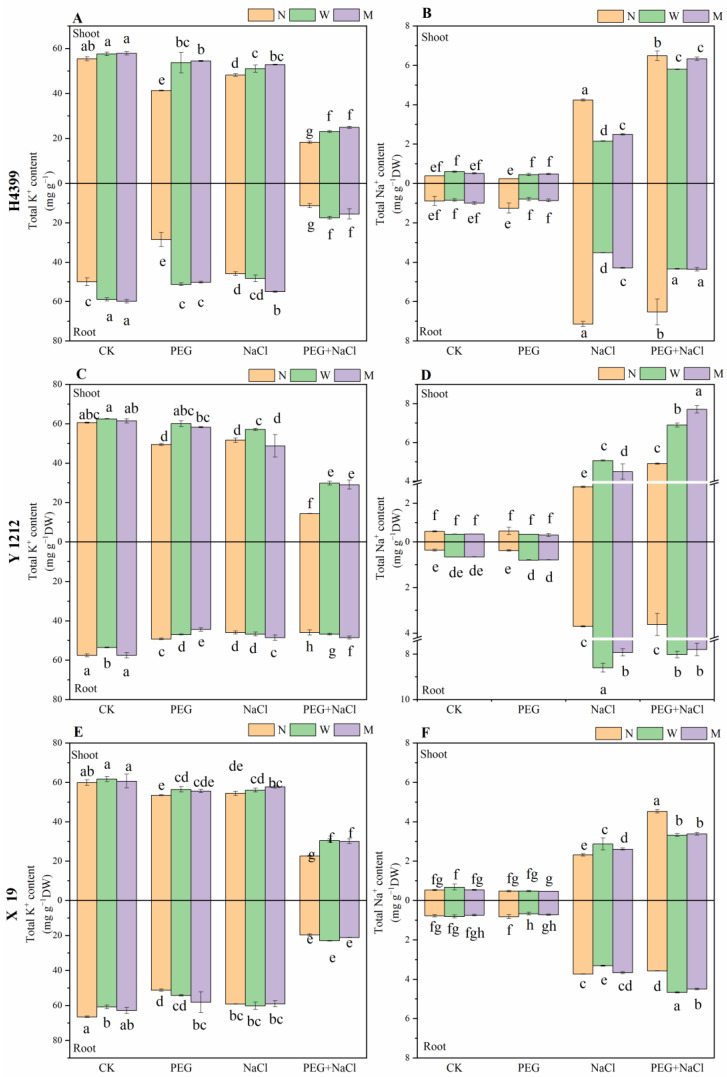
The K^+^ and Na^+^ content in shoot and root of three wheat seedlings from no priming (N), water priming (W), and melatonin priming (M) under PEG, NaCl, and PEG + NaCl treatments. (**A**,**B**) represent the parameters in H4399; (**C**,**D**) represent the parameters in Y1212; (**E**,**F**) represent the parameters in X19. The standard deviation (SD) value from three replicates is represented as error bars in figures, and different lowercase letters indicate significant differences at the 0.05 level.

**Figure 7 ijms-25-05055-f007:**
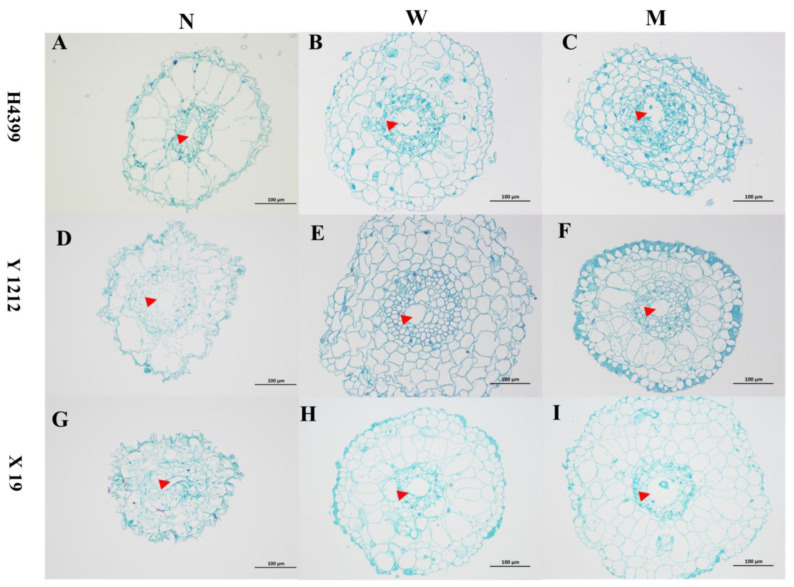
The observation of root cross section in three wheat seedlings from no priming (N), water priming (W), and melatonin priming (M) under PEG + NaCl treatment. (**A**–**C**) represent the H4399; (**D**–**F**) represent the Y1212; (**G**–**I**) represent the X19; (**A**,**D**,**G**) represent the no-priming (N) treatment; (**B**,**E**,**H**) represent the water-priming (W) treatment; (**C**,**F**,**I**) represent the melatonin-priming (M) treatment. The red triangle indicates the root canal tissue.

**Figure 8 ijms-25-05055-f008:**
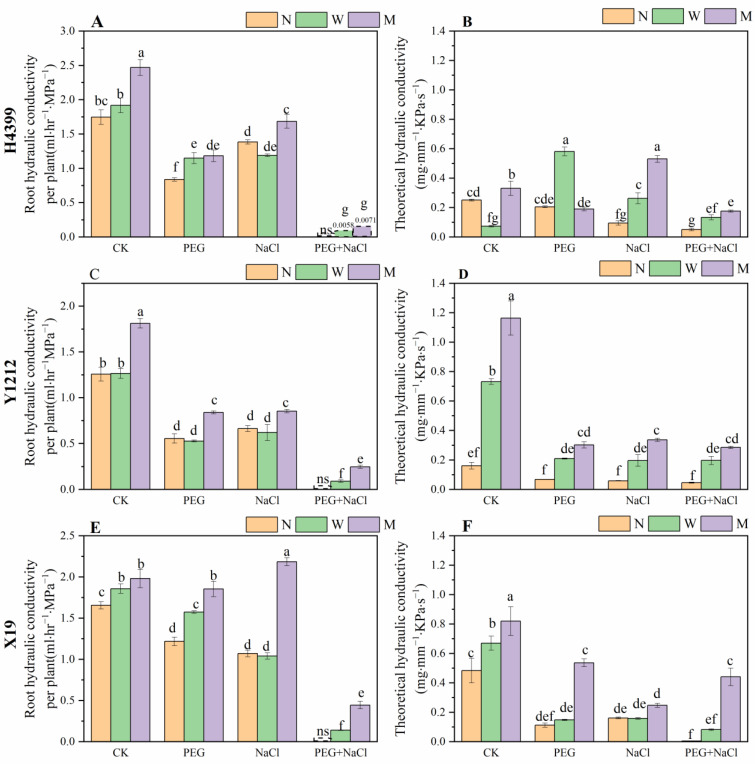
The root hydraulic conductivity and theoretical hydraulic conductivity of three wheat seedlings from no priming (N), water priming (W), and melatonin priming (M) under PEG, NaCl, and PEG + NaCl treatments. (**A**,**B**) represent the parameters in H4399; (**C**,**D**) represent the parameters in Y1212; (**E**,**F**) represent the parameters in X19. The standard deviation (SD) value from three replicates is represented as error bars in figures, and different lowercase letters indicate significant differences at the 0.05 level.

**Figure 9 ijms-25-05055-f009:**
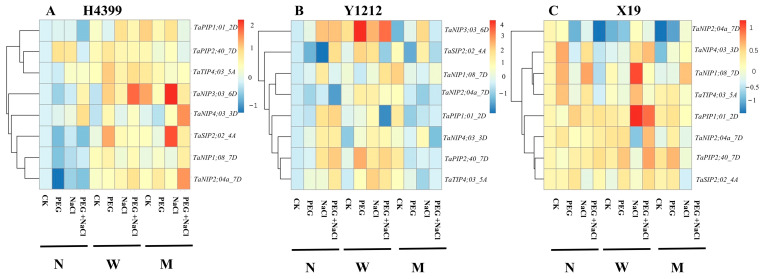
The heat map analysis of some aquaporin genes in root of three wheat seedlings from no priming (N), water priming (W), and melatonin priming (M) under PEG, NaCl, and PEG + NaCl treatments.

**Figure 10 ijms-25-05055-f010:**
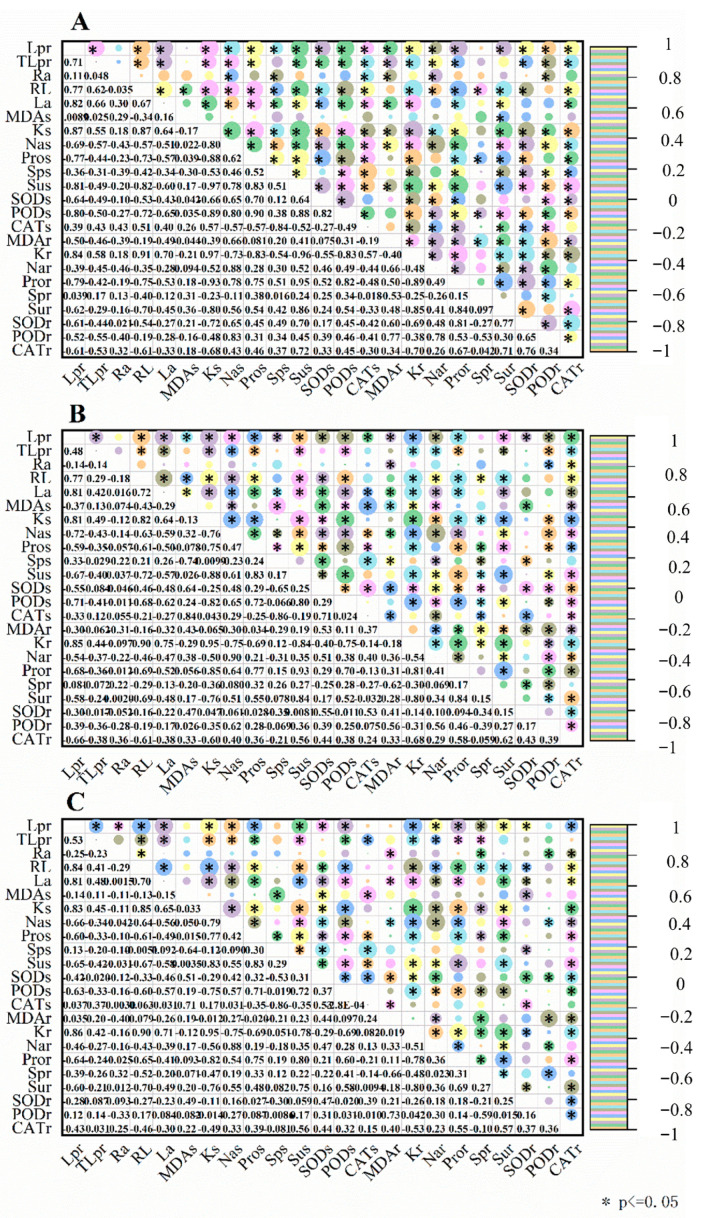
The Pearson correlation analysis between the hydraulic conductivity and physiological parameters measured from no priming (**A**), water priming (**B**), and melatonin priming (**C**) under PEG, NaCl, and PEG + NaCl treatments. Lpr = root hydraulic conductivity per plant; TLpr = theoretical hydraulic conductivity; Ra = root activity; RL = root length; La = leaf area; MDAs= MDA content in shoot; MDAr = MDA content in root; Ks = K^+^ content in shoot; Kr = K^+^ content in root; Nas = Na^+^ content in shoot; Nar = Na^+^ content in root; Pros = proline content in shoot; Pror = proline content in root; Sps = soluble protein content in shoot; Spr = soluble protein content in root; Sus = soluble sugar content in shoot; Sur = soluble sugar content in root; SODr = SOD activity in root; SODs = SOD activity in shoot; PODs = POD activity in shoot; PODr = POD activity in root; CATs = CAT activity in shoot; CATr = CAT activity in root; * indicates a significant difference at the level of *P*<0.05.

**Figure 11 ijms-25-05055-f011:**
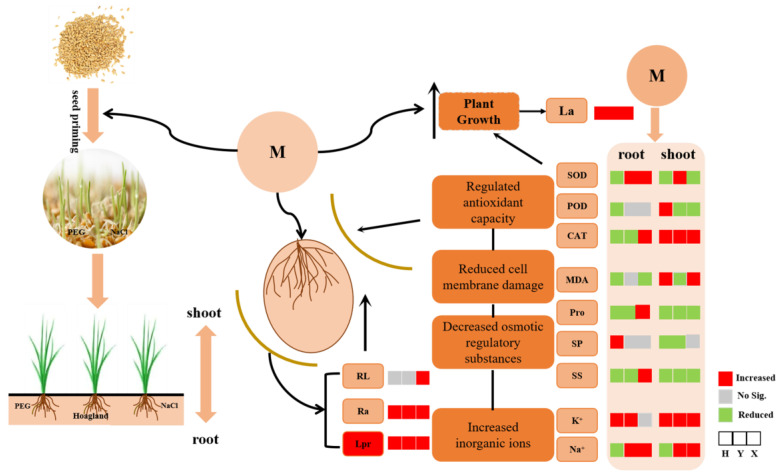
Proposed mechanism of action underlying the enhancement of seedling growth by melatonin priming under PEG + NaCl stress. Lpr = root hydraulic conductivity per plant; RL = root length; Ra = root activity; La = leaf area; Pro = proline content; SP = soluble protein content; SS = soluble sugar content. M = melatonin priming; W = water priming; H = Hengmai 4399; Y = Yannong 1212; X = Xinmai 19.

**Table 1 ijms-25-05055-t001:** The antioxidant enzyme in shoots and roots of three wheat seedlings from no priming (N), water priming (W), and melatonin priming (M) under PEG, NaCl, and PEG + NaCl treatments.

			SOD		POD		CAT	
			(U mg^−1^ prot^−1^ DW)		(U g^−1^ prot^−1^ DW)		(μmol^−1^min^−1^ mg^−1^ prot^−1^ DW)	
			Shoot	Root	Shoot	Root	Shoot	Root
H 4399	CK	N	50.85 ± 2.89 d	47.58 ± 1.83 ef	0.93 ± 0.10 e	16.15 ± 0.24 fg	2.45 ± 0.10 de	2.65 ± 0.05 e
		W	31.45 ± 0.56 ef	56.36 ± 1.20 ef	1.46 ± 0.14 cd	23.38 ± 0.75 de	2.27 ± 0.13 e	2.64 ± 0.32 e
		M	31.07 ± 0.17 ef	82.96 ± 3.41 e	1.02 ± 0.11 de	26.24 ± 3.41 cd	2.35 ± 0.11 e	4.39 ± 0.23 d
	PEG	N	53.85 ± 1.59 d	149.60 ± 9.03 d	1.32 ± 0.05 cde	15.18 ± 0.39 g	3.28 ± 0.10 c	8.29 ± 0.22 b
		W	27.43 ± 4.73 f	342.25 ± 17.64 b	1.08 ± 0.07 de	13.39 ± 0.60 g	2.32 ± 0.05 e	4.47 ± 0.26 d
		M	30.68 ± 0.69 ef	120.85 ± 6.21 d	1.20 ± 0.02 cde	15.01 ± 0.50 g	2.73 ± 0.26 cde	4.97 ± 0.25 d
	NaCl	N	75.12 ± 1.17 b	187.93 ± 12.61 c	1.42 ± 0.06 cde	39.31 ± 0.95 b	2.32 ± 0.15 e	7.74 ± 0.81 bc
		W	37.27 ± 1.68 e	76.90 ± 1.27 e	1.63 ± 0.11 c	19.83 ± 0.88 ef	2.92 ± 0.07 cd	0.83 ± 0.13 f
		M	49.16 ± 0.89 d	149.05 ± 5.36 d	1.18 ± 0.02 cde	26.11 ± 1.36 cd	3.22 ± 0.01 c	2.58 ± 0.21 e
	PEG + NaCl	N	94.89 ± 2.09 a	437.52 ± 35.16 a	4.92 ± 0.16 b	44.72 ± 0.84 a	3.03 ± 0.17 c	11.91 ± 0.34 a
		W	69.65 ± 3.72 bc	213.28 ± 7.70 c	5.80 ± 0.39 a	29.14 ± 2.53 c	5.65 ± 0.38 a	8.81 ± 0.49 b
		M	65.88 ± 3.85 c	18.85 ± 0.48 f	6.13 ± 0.22 a	17.02 ± 0.60 fg	4.87 ± 0.19 b	6.78 ± 0.15 c
Y 1212	CK	N	54.88 ± 1.49 f	104.14 ± 1.12 g	1.23 ± 0.10 de	30.35 ± 0.58 de	3.12 ± 0.06 f	5.99 ± 0.62 de
		W	124.94 ± 1.84 b	362.57 ± 5.85 f	0.79 ± 0.05 g	29.14 ± 0.41 e	4.87 ± 0.08 e	5.24 ± 0.93 ef
		M	102.24 ± 0.40 c	458.96 ± 9.00 e	1.83 ± 0.11 b	46.52 ± 3.26 a	4.94 ± 0.06 de	10.76 ± 1.26 ab
	PEG	N	74.60 ± 1.39 e	160.98 ± 11.43 g	0.92 ± 0.04 fg	27.25 ± 1.36 e	3.15 ± 0.04 f	8.91 ± 1.06 bc
		W	124.17 ± 1.46 b	2656.57 ± 42.03 a	1.61 ± 0.03 c	29.70 ± 1.00 de	5.30 ± 0.24 cd	10.92 ± 1.08 ab
		M	103.34 ± 1.40 c	1717.18 ± 47.65 b	1.88 ± 0.02 b	28.73 ± 0.35 e	4.60 ± 0.01 e	7.44 ± 0.56 cde
	NaCl	N	49.21 ± 7.96 f	137.01 ± 1.56 g	1.57 ± 0.03 c	34.24 ± 0.54 c	2.63 ± 0.19 g	3.35 ± 0.64 f
		W	147.93 ± 3.02 a	902.13 ± 33.12 cd	2.01 ± 0.08 b	32.82 ± 0.45 cd	7.05 ± 0.17 a	6.63 ± 0.33 cde
		M	145.31 ± 1.69 a	520.22 ± 37.91 e	1.34 ± 0.04 d	29.56 ± 0.30 de	5.38 ± 0.23 c	7.95 ± 0.46 cd
	PEG + NaCl	N	76.97 ± 3.79 e	532.12 ± 12.55 e	4.34 ± 0.16 a	42.97 ± 0.56 b	1.86 ± 0.03 h	12.35 ± 0.45 a
		W	142.15 ± 0.49 a	852.59 ± 17.53 d	1.04 ± 0.05 ef	40.00 ± 1.12 b	5.98 ± 0.10 b	10.57 ± 0.48 ab
		M	87.55 ± 0.87 d	936.84 ± 22.3 c	0.51 ± 0.01 h	40.83 ± 0.09 b	3.43 ± 0.04 f	9.05 ± 0.53 bc
X 19	CK	N	57.35 ± 1.72 d	63.92 ± 0.94 g	1.09 ± 0.08 e	21.64 ± 0.15 f	3.44 ± 0.05 b	2.65 ± 0.61 fg
		W	39.33 ± 1.59 f	102.09 ± 3.24 efg	0.91 ± 0.04 ef	31.12 ± 3.19 cde	2.60 ± 0.09 d	3.89 ± 0.12 f
		M	49.58 ± 0.62 e	155.56 ± 4.62 d	0.60 ± 0.02 fg	39.70 ± 4.72 b	2.65 ± 0.09 cd	5.80 ± 0.48 e
	PEG	N	60.26 ± 1.01 d	212.85 ± 17.77 c	1.03 ± 0.01 e	25.39 ± 0.67 ef	3.46 ± 0.10 b	9.63 ± 0.31 b
		W	48.86 ± 1.06 e	483.80 ± 12.20 a	0.32 ± 0.04 g	37.87 ± 1.06 bc	2.49 ± 0.01 d	8.68 ± 0.17 bc
		M	38.67 ± 1.65 f	160.99 ± 20.26 d	0.66 ± 0.05 fg	33.46 ± 2.40 bcd	2.58 ± 0.10 d	5.81 ± 0.46 e
	NaCl	N	66.73 ± 1.12 c	150.79 ± 3.98 de	1.02 ± 0.06 e	37.50 ± 0.44 bcd	3.33 ± 0.18 b	1.96 ± 0.52 g
		W	48.95 ± 0.89 e	104.14 ± 2.95 efg	0.87 ± 0.04 ef	35.43 ± 0.63 bcd	2.91 ± 0.02 c	3.89 ± 0.51 f
		M	71.74 ± 0.95 c	119.81 ± 4.76 def	1.70 ± 0.17 d	61.48 ± 1.67 a	3.58 ± 0.11 b	7.63 ± 1.12 cd
	PEG + NaCl	N	145.96 ± 3.51 a	69.60 ± 0.46 fg	7.02 ± 0.19 a	31.17 ± 0.40 cde	2.94 ± 0.13 c	6.65 ± 0.12 de
		W	89.85 ± 1.51 b	346.12 ± 36.88 b	3.14 ± 0.14 b	30.57 ± 0.88 de	4.29 ± 0.03 a	13.78 ± 0.01 a
		M	66.03 ± 4.49 c	496.57 ± 33.80a	2.17 ± 0.24 c	25.46 ± 2.77 ef	4.01 ± 0.07 a	9.26 ± 0.04 b

Note: The standard deviation (SD) value from three replicates is represented as error bars in figures, and different lowercase letters indicate significant differences at the 0.05 level.

## Data Availability

All data presented in the study are included in the article. Further inquiries can be directed to the corresponding author.

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
