# Peer review of "Seeds Priming with Melatonin Improves Root Hydraulic Conductivity of Wheat Varieties under Drought, Salinity, and Combined Stress"

_ijms, 2024, doi:10.3390/ijms25095055_

Round 1
Reviewer 1 Report
Comments and Suggestions for Authors
The present manuscript “Seed priming with melatonin improves root hydraulic conductivity of wheat varieties under drought, salinity, and combined stress” is an interesting idea. The manuscript contains novel findings that will provide new insights regarding the use of melatonin to mitigate drought and salinity stress in plants. I have some concerns/suggestions that authors can incorporate and I believe this will improve the quality of their work.
In abstract section lines 14-16: These lines must be revised as they are not given any clear message.
Lines 25-25: The present study suggested that melatonin-seed priming was a strategy as regards the enhancement of root hydraulic conductivity under PEG, NaCl, and PEG+NaCl stress, leading to efficiently improve wheat resistant to drought-salinity stress.
The quality of the English language in these lines is very poor and it is misleading the reader. Therefore, authors must revise these lines to give a clear message to readers.
The abstract section is very poorly written in a very descriptive form, and it can be enriched by adding some numerical values of study findings.
The keywords are just the repetition of the title and they must be changed and should be different from the title.
The first sentence of the introduction: (Drought and salinity stress are two major causes for limited cereal crops development, growth, and quality from seed germination to harvest in many regions of world, especially for wheat cultivation (Alam et al., 2022; Isayenkov, 2019; Wang et al., 2018). The quality English language is quite awkward here, therefore, I suggest the authors revise these sentences.
The introduction section should be written in state-of-the-art form, and there is a dire need to improve the readability of the text from the abstract to the way conclusion.
Introduction should be divided into different parts: 1) problem statement, 2) effects of drought on plants, 3) effects of salinity on plants, 4) combined effects of drought and salinity on plants, 5) effect of melatonin on plants under drought and salinity and how melatonin can mitigate drought and salinity toxicity, 6) importance of wheat, 7) clear hypothesis, and objectives.
The varieties [(Heng4399 (H4399), Yannong1212 (Y1212), and Xinmai19(X19)] were drought and salt tolerant or sensitive?
Line 137: The seedlings were hydroponic with Hoagland solution???
Detailed information should be provided on the detection of the physiological index in the revised version.
The statistical section should be self-explanatory. How data was analyzed? And how means were separated?
Please avoid stating results from Figure 1, relative to CK, most of PEG, NaCl, and PEG+NaCl stresses obviously.
The results section is very long and contains much unnecessary information, therefore, authors must revise this section and try to be precise and concise in this section.
The title of section 4.1 (seed-priming using melatonin improving root hydraulic conductivity) should be revised.
The discussion section must be carefully revised it is full of study results which are not needed here.
I have a serious concern on the discussion section, it should be enriched with logical reasoning and supported with study findings rather than general statements.
There are also some mistakes in English grammar, syntax, and punctuation that distract the readers. Please carefully review your manuscript before re-submission for the quality of English.
Comments on the Quality of English Language
The manuscript needs English editing from editing services.
Author Response
|
Response to Reviewer 1Comments
|
||
|
1. Summary |
|
|
|
Thank you very much for taking the time to review this manuscript. Please find the detailed responses below and the corresponding revisions in the re-submitted files.
|
||
|
2. Questions for General Evaluation |
Reviewer’s Evaluation |
Response and Revisions |
|
Does the introduction provide sufficient background and include all relevant references? |
Must be improved |
Revisions |
|
Are all the cited references relevant to the research? |
Yes |
|
|
Is the research design appropriate? |
Yes |
|
|
Are the methods adequately described? |
Must be improved |
Revisions |
|
Are the results clearly presented? |
Can be improved |
Revisions |
|
Are the conclusions supported by the results? |
Can be improved |
|
|
3. Point-by-point response to Comments and Suggestions for Authors |
||
|
Comments 1: In abstract section lines 14-16: These lines must be revised as they are not given any clear message. Lines 25-25: The present study suggested that melatonin-seed priming was a strategy as regards the enhancement of root hydraulic conductivity under PEG, NaCl, and PEG+NaCl stress, leading to efficiently improve wheat resistant to drought-salinity stress. The quality of the English language in these lines is very poor and it is misleading the reader. Therefore, authors must revise these lines to give a clear message to readers.
|
||
|
Response 1: Thanks for pointing out the terrific and very important question. We had made corresponding revisions in the manuscript (Line #15-18; Line #29-31). |
||
|
Comments 2: The abstract section is very poorly written in a very descriptive form, and it can be enriched by adding some numerical values of study findings. |
||
|
Response 2: We highly appreciate the illuminating and valuable comment and suggestions given by the reviewer, which has helped us to greatly improve the quality of our manuscript. Accordingly, we have made corresponding revisions in the manuscript (Line #22-27). |
||
|
Comments 3:The keywords are just the repetition of the title and they must be changed and should be different from the title. |
||
|
Response 3:Thanks. We had amended the part accordingly (Line #32). |
||
|
Comments 4:The first sentence of the introduction: (Drought and salinity stress are two major causes for limited cereal crops development, growth, and quality from seed germination to harvest in many regions of world, especially for wheat cultivation (Alam et al., 2022; Isayenkov, 2019; Wang et al., 2018). The quality English language is quite awkward here, therefore, I suggest the authors revise these sentences. |
||
|
Response4: We thank the reviewer for pointing out the important point.
We have revised the sentence in the part accordingly(Lines #35-38). |
||
|
Comments 5:The introduction section should be written in state-of-the-art form, and there is a dire need to improve the readability of the text from the abstract to the way conclusion.
Introduction should be divided into different parts: 1) problem statement, 2) effects of drought on plants, 3) effects of salinity on plants, 4) combined effects of drought and salinity on plants, 5) effect of melatonin on plants under drought and salinity and how melatonin can mitigate drought and salinity toxicity, 6) importance of wheat, 7) clear hypothesis, and objectives.
|
||
|
Response 5:We thank the reviewer for pointing out the important point and giving the good suggestion. We had adjusted the structure of ‘Introduction’ in our manuscript (Line #35-137). |
||
|
Comments 6:The varieties [(Heng4399 (H4399), Yannong1212 (Y1212), and Xinmai19(X19)] were drought and salt tolerant or sensitive? |
||
|
Response 6: Thanks. Heng4399 (H4399) was resistance to drought and salt stress. Yannong1212 (Y1212) was moderately resistance to drought stress and sensitive to salt stress while Xinmai19 (X19) was sensitive to drought and salt stress. Accordingly, we have made corresponding revisions in the manuscript (Line #131-133). |
||
|
Comments 7: Line 137: The seedlings were hydroponic with Hoagland solution??? |
||
|
Response 7: Thanks. We have made revisions in the corresponding places of the manuscript (Line #148). |
||
|
Comments 8: Detailed information should be provided on the detection of the physiological index in the revised version. The statistical section should be self-explanatory. How data was analyzed? And how means were separated? |
||
|
Response 8: We thank the reviewer for giving the good suggestion. We have gladly accepted the suggestion and revised the according manuscript (Line # 157-162;Line # 206-222). |
||
|
Comments 9: Please avoid stating results from Figure 1, relative to CK, most of PEG, NaCl, and PEG+NaCl stresses obviously. |
||
|
Response 9: We thank the reviewer for the valuable suggestion. We have gladly accepted the suggestion and rechecked this point carefully throughout the manuscript. Accordingly, we have deleted these and revised corresponding manuscript (Line#240;Line#260 ). |
||
|
Comments 10: The results section is very long and contains much unnecessary information, therefore, authors must revise this section and try to be precise and concise in this section.
The title of section 4.1 (seed-priming using melatonin improving root hydraulic conductivity) should be revised. |
||
|
Response 10: Thanks. We have deleted the corresponding results and revised the discussion in the manuscript (Line #239-648ï¼›Line #656 ). |
||
|
Comments 11: The discussion section must be carefully revised it is full of study results which are not needed here. I have a serious concern on the discussion section, it should be enriched with logical reasoning and supported with study findings rather than general statements.
|
||
|
Response 11: We thank the reviewer for the important comment. We have revised the corresponding discussion in the manuscript (Line #655-768). |
||
|
4. Response to Comments on the Quality of English Language |
||
|
Point 1: There are also some mistakes in English grammar, syntax, and punctuation that distract the readers. Please carefully review your manuscript before re-submission for the quality of English. |
||
|
Response 1: Thanks. We had reviewed the proper English language, grammar, punctuation, spelling, and overall style of the entire manuscript to give a clear message to readers. |
||

Reviewer 2 Report
Comments and Suggestions for Authors
The current paper devoted to investigation the effect of melatonine as seeds priming agent on wheat seedlings/plants resistance to stress. Authors have proved that priming significantly improve root hydrailic conductivity and other plant parameters under PEG and NaCl.
The results are very interesting and important from the scientific and practical point of view. However, there are so many unclarity in results presentation. In discussion part role of epigenetic memory is missing. Actualy, seeds priming have a major effcet on epigenetic memory and on synhronization of organ establishment. This point must be add to discussion.
However, the paper need significant corrections.
Below are some essentail details need to be corrected:
For example, details of seedling growth are missing (volume oft he medium, type of the bottle etc).
Parametrs measurements are really confusing (lines 153 – 162). There are many protocols in one part. What is root activity?
Lines 169 – 182_ very confused part. Plesae, make an order with punctuation, separate protocols , give explanations.
„physiological index“ ??
QRT-PCR = qPCR. What is TAction gene? Maybe Actin gene? Please, provide link to the promer here.
Figure 1: it seems only H2349 have a most positive effect on proming. Can you provide some „speculation“ why? Any special traits reklated with this line?
Line 257: “enhanced root structure” ¿? Structure can not enhanced!!!
Part 3.2: root activity?
Figure 2-4: all parametrs were measured per FW, what is not scientific. Under PEG/NaCl cell are smaller because of leass water contents. The best way is to give results per cellular base (per cell). Please, mention this points and try to explain.
Table 1: „the antionxidant wnzyme“ ???? What is plot? Please, clarify units. It should be per protein.
Figure 7: it is better to provide more contrast images: white-blue on gray background is not very visible. If possible, make section black or at least more dark.
Title: Seed = seeds.
Line 11-12: in will be better to mention here time of melatonin applications (priming).
Line 14: „three wheat seed“ = seeds of thres wheat lines“.
Line 16: „ H4399, Y1212, and X19“ – this number stell us nothing. It is better to mention wheat „type“ here.
„osmosis substances“ ??
Line 40: „increase of membrane osmotic regulator”??
Line 98: 2021)2021 ¿?
Line 122: please, provide full description: Triticum aestivum cultivars? Etc.?
Line 133: concentration = µM, not µmol. µmol is amount.
Line 622: “Recently, drought stress, salt stress, and dual drought-salt stress obviously reduced Lpr“ ??? Maybe „recently it was shown“?
Comments on the Quality of English Language
The english is really a problem in the current version. There are so many mistakes. Some I were mentoined, but there are much more..
Author Response

(The authors gave the same response as above.)

Round 2
Reviewer 1 Report
Comments and Suggestions for Authors
NA
Comments on the Quality of English LanguageMinor changes are needed.
Author Response
Thanks you very much for reviewing this manuscript!
Reviewer 2 Report
Comments and Suggestions for Authors
Thank you! The text is much better now. Small corrections still required.
Lines 173 – 175: these statements are confucng. Please, change the order.
Lin 612: “Under NaCl stress, there was no different among three priming
treatments in the content of H4399 by comparison.” ¿??
Table 1: please, clarify units.
Lines 1094 – 1098: it is rather viceversa, indeed. Higher stress tolerance may be results of specific epigenetics etc.
Comments on the Quality of English Languagesome polishing during proof reading
Author Response
|
Response to Reviewer 2 Comments
|
||
|
1. Summary |
|
|
|
Thank you very much for taking the time to review this manuscript. Please find the detailed responses below and the corresponding revisions in the re-submitted files.
|
||
|
2. Questions for General Evaluation |
Reviewer’s Evaluation |
Response and Revisions |
|
Does the introduction provide sufficient background and include all relevant references? |
Yes |
|
|
Are all the cited references relevant to the research? |
Yes |
|
|
Is the research design appropriate? |
Yes |
|
|
Are the methods adequately described? |
Yes |
|
|
Are the results clearly presented? |
Yes |
|
|
Are the conclusions supported by the results? |
Yes |
|
|
3. Point-by-point response to Comments and Suggestions for Authors |
||
|
Comments 1: Lines 173 – 175: these statements are confucng. Please, change the order. |
||
|
Response 1: Thank you for pointing this out. We had revised these statements (Line #173-176). |
||
|
Comments 2: Lin 612: “Under NaCl stress, there was no different among three priming treatments in the content of H4399 by comparison.” ¿?? |
||
|
Response 2: Thanks for pointing out the terrific and very important question. Agree. We had revised it in the corresponding manuscript (Line #355). |
||
|
Comments 3:Table 1: please, clarify units. |
||
|
Response 3: Agree. We had, accordingly, modified the units with ‘U mg-1 prot-1 DW’ and ‘μmol-1min-1 mg-1 prot-1 DW’ in the corresponding manuscript (Line #445). |
||
|
Comments 4: Lines 1094 – 1098: it is rather viceversa, indeed. Higher stress tolerance may be results of specific epigenetics etc. |
||
|
Response 4: We thank the reviewer for pointing out the important point and giving the good suggestion. We had revised the corresponding places in the manuscript (Line #761). |
||
